# Continental scale dietary patterns in a New World raptor using web-sourced photographs

**Connor T. Panter**[1]*, **Vincent N. Naude**[2], **Facundo Barbar**[3], **Arjun Amar**[4]

**1** School of Geography, University of Nottingham, Nottingham, United Kingdom, **2** Department of Conservation Ecology and Entomology, University of Stellenbosch, Matieland, South Africa, **3** Laboratorio Ecotono INIBIOMA (CONICET-Universidad Nacional del Comahue), San Carlos de Bariloche, Río Negro, Argentina, **4** FitzPatrick Institute of African Ornithology, University of Cape Town, Cape Town, South Africa

* connor.panter@nottingham.ac.uk, connorpanter1301@gmail.com

**Data Availability Statement:** All relevant data for this study are publicly available from the Dryad repository (https://doi.org/10.5061/dryad.rjdfn2zkz).

## Abstract

Dietary studies are essential to better understand raptor ecology and resource requirements through time and space, informing species habitat use, interspecific interactions and demographic rates. Methods used to collect data on raptor diets can constrain how dietary analyses can be interpreted. Traditional approaches to study raptor diets, such as analysis of pellets or prey remains, often provide dietary data at the local population level and tend to be restricted to pairs during the breeding season. The increasing use of citizen science data has the potential to provide dietary inferences at larger spatial, demographic and temporal scales. Using web-sourced photography, we explore continental-scale demographic and latitudinal dietary patterns between adult and non-adult Crested Caracaras (*Caracara plancus*), throughout the species' range across the Americas. We analysed 1,555 photographs of caracaras feeding and found no age effects on the probabilities of different food groups being included in photographs. The probability of reptiles being included in photographs of caracaras from the northern population was significantly higher than those from the southern population, with the opposite pattern for birds. There were significant latitudinal effects with the probabilities of fishes and invertebrates in the diet of northern caracaras increasing towards the equator. Contrastingly, the probability of mammals in the diet increased away from the equator for both populations. Assuming the focal species is well-sampled, web-sourced photography can improve our understanding of raptor diets at large-scales and complements more traditional approaches. This approach is more accessible to raptor researchers without access to the field or expertise in physical prey identification techniques.

## Introduction

Dietary studies are essential to better understand a species' ecology and resource requirements through time and space [1]. Food availability and abundance can affect population dynamics of raptors [2]. Understanding raptor diets provides a foundation to build upon knowledge about habitat use [3], inter- and intraspecific interactions [4], demographic rates [5], threats

**Funding:** The author(s) received no specific funding for this work.

**Competing interests:** The authors have declared that no competing interests exist.

including direct consumption and bioaccumulation of contaminants [6, 7], and other aspects of raptor ecology [8]. For generalist raptor species that often scavenge on animal carcasses, dietary information may benefit humans in the form of monitoring ecosystem service provisioning [9]. Most importantly, the methods used to collect data on raptor diets, and the scales at which data are collected at, may lead to a misunderstanding of natural environments due to the introduction and constraints of methodological limitations.

Traditional methods to study raptor diets, such as analysis of prey remains [10, 11], pellets [12], observations from bird blinds [13] and use of nest cameras [10, 14], may be restricted spatially in their ability to provide dietary data at the broader population-level. Furthermore, such approaches tend to often provide dietary data at the local population-level and are limited to pairs during the breeding season. Therefore, these approaches tend not to capture the diets of floaters, sexually immature birds or those that fail to breed [15]. More recent methods to explore raptor diet include coupling accelerometers to telemetry devices [16], stable isotope analyses [17–19] and DNA metabarcoding [4, 20, 21]. However, when attempting to explore dietary patterns at the wider population-level across larger scales, these approaches tend to be time- and cost-inefficient.

The increasing popularity and use of open-source intelligence and citizen science data [22], in the form of web-sourced photographs, has the potential to provide dietary inferences at larger spatial scales [23], complementing more traditional approaches. For species that inhabitant large geographical areas, exploring how diet varies latitudinally may improve understanding of its ecology. For example, ecological and climatic conditions along latitudinal gradients influence the availability and presence of prey species within the wider environment [24], affecting the diversity and dietary composition for many species including birds [25]. Reviews of the raptor diet literature found distinct latitudinal patterns in the proportions of prey groups in some raptor diets, with increased probabilities of mammalian prey at higher latitudes [25, 26]. In addition to these studies, web-sourced photography has already been used to explore raptor diets across large geographic areas. For example, it has been used to examine spatial differences in the diet of adult and non-adult Martial Eagles (*Polemaetus bellicosus*) throughout sub-Saharan Africa [27]. Studies have also explored sex- and age-related diet differences in the Eurasian Sparrowhawk (*Accipiter nisus*) and explored changes in this species' diet, between the sexes, throughout the entire year [28, 29]. Previous research, using this approach, examined diet specialization of Tiny Hawks (*A. superciliosus*) in relation to their hummingbird (Trochilidae) prey across the Neotropics [30]. However, the role of more traditional approaches to study raptor diets must not be underestimated as they provide vital data on cryptic species and those that can be challenging to study in the field, e.g., tropical forest raptors [31]. Despite this, web-sourced photography is a useful tool to study well-sampled species that inhabitant large geographic areas including open landscapes and those associated with human activities [28, 29].

The Crested Caracara (*Caracara plancus*) (hereafter 'caracara') is distributed throughout the Americas. It is a non-migratory, resident species that tends to occupy distinct territories outside of the breeding season [32]. The species' most northerly limit extends across breeding populations in Texas [33], southern Arizona [34] and Florida [35–37]. The caracara's southernmost range extends to the Tierra del Fuego archipelago in South America [38]. Dietary research has been conducted at the local population-levels (i.e., Florida, USA [37, 39–42]; Texas, USA [43]; Argentina [44–47]; Brazil [48]), however, little is known about the species' diet at the continental-scale and how this may vary along a latitudinal gradient. Furthermore, how diet differs between age groups remains unknown.

To address this information gap, we used web-sourced photographs accessed from Macaulay Library and iNaturalist of caracaras feeding and examined spatial and demographic

differences in the diet throughout the species' range across the Americas. We employed a single technique to compare diet across a broad range of the species' global population and provide the first dietary assessment between age groups. Specifically, we explore 1) dietary differences between adult and non-adult birds, 2) differences in probabilities of food groups between northern and southern populations, and 3) examine how the probabilities of different food groups within the caracaras' diet differ along a latitudinal gradient. In line with previous research on the effects of latitude on raptor diet, we expect a distinct latitudinal effect on the probability of mammals within the caracara diet—consistent with known trends in mammalian prey assemblages [26], i.e., an increased probability of mammalian prey across northern latitudes [49] and towards the poles [25]. In addition, due to global climatic and species diversity patterns varying along latitudinal gradients spanning temperate through tropical biomes [50–52], we expect the probability of ectothermic food groups, e.g., fishes, invertebrates and reptiles, to increase in the caracaras' diet towards the equator.

## Materials and methods

### Study areas

Our study area encompasses the caracara's entire geographic range, spanning a latitudinal gradient from approximately 45˚N to -55˚S, and a longitudinal gradient from approximately -35˚E to -125˚W. Photographic samples span multiple regions including the Nearctic of North America, Mexican Transition Zone through North/Central America and into the Neotropical zone in Central/South America [53]. According to the Bioregions 2023 Framework (https://www.oneearth.org/bioregions/), North American photographs represent samples from numerous biogeographic subrealms including the North Pacific Coast, American West, Mexican Drylands, Great Plains and the Southeast U.S. Savannas and Forests. Photographs of caracaras from Central America include the Central America and Caribbean biogeographic subrealms, with those from South America spanning the Andes and Pacific Coast, Upper South America, Amazonia, Brazilian Cerrado and Atlantic Coast and the South American Grassland subrealms.

### Web-sourced photographs and data extraction

**Macaulay Library and iNaturalist data.** All photographs and available metadata of 'Crested Caracaras (*Caracara plancus*)' uploaded onto Macaulay Library (https://www.macaulaylibrary.org/) and iNaturalist (https://www.inaturalist.org./) were downloaded on 27 January 2023 and 2 February 2024, respectively, where the use of these images for non-commercial scientific research purposes fell under the CC BY 4.0 license as authors were acknowledged where relevant in compliance with the terms and conditions for the sources of these data (i.e., Macaulay Library—https://www.birds.cornell.edu/home/ebird-data-access-terms-of-use/ and iNaturalist—https://www.inaturalist.org/pages/terms). Photographs were visually examined and classified as 'of interest' if they appeared to include caracaras hunting or scavenging on food items. The 'of interest' photographs were filtered again by a second researcher who confirmed whether the photograph contained a caracara either hunting or scavenging. These photographs were then classified as 'useable'. For them, we extracted the following data: 1) caracara age (adult vs. non-adult), 2) food item (identified to the lowest taxonomic level possible), 3) latitude/longitude, 4) location of photograph and 5) observation date. For photographs that contained more than one caracara of each age group, we only extracted data for the individual interacting with the prey item. We checked for duplicated photographs of the same feeding events by filtering by observation date, visual characteristics or landscape features within the photograph and Macaulay Library recordist name or iNaturalist user ID, i.e., a

duplicated photograph was identified if the photograph was taken within the same area, within up to 10 days of the initial observation and if the same recordist uploaded the photo. However, this was not always the case as there were occasions where multiple recordists uploaded images of the same feeding event across different days. This was particularly the case if the food item was large in size and thus took longer to decompose. We were able to ascertain whether these were duplicates by making an assessment based on the spatial location of the feeding event, evidence of the food item included and features of the surrounding location in the photograph. If a duplicated photograph was identified, then we retained the highest resolution photograph that showed the feeding event most clearly and removed all others.

### Prey identification and caracara age

We attempted to identify food items to the lowest taxonomic level possible. We use the term 'food item' here as we were unable to determine whether observations reflected feeding events where caracaras actively hunted/killed the item or were photographed scavenging on carcasses not killed by the bird. Where we were unable to identify food items to species-level, we assigned these to a broader food group, e.g., 'birds' or 'mammals'. If we could not identify a food item to a broader 'food group', e.g., a photograph of a partially decomposed body part or unidentifiable bone, it was categorized as 'unknown'. All photographs of species- and food group-level identifications were visually assessed by at least two of the authors (C.P., F.B. or V.N.).

Adult caracaras were identified by characteristic dark plumage, whitish-buff auricular feathers, throat and nape, and whitish-buff barred dark chest, neck, mantle, back, upper tail coverts, crissum and basal part of the tail [54]. Juvenile birds tend to resemble adults in pattern, but are paler brown with vertical streaking on the chest, neck and back, also displaying grey legs and lighter ceres [54]. We classified all individuals that did not display adult characteristics as 'non-adults'.

### Population classification

Until recently, two distinct subspecies of caracara were recognized, i.e., the Northern Crested Caracara (*C. p. cheriway*) and the Southern Crested Caracara (*C. p. plancus*) [55]. The northern subspecies' geographic distribution extends from Texas, southern Arizona and Florida, down to northern Amazonian rainforest into Brazil and Peru [33]. The southern subspecies' geographic distribution extended from north-east Brazil, Bolivia and Chile down to the Tierra del Fuego archipelago in South America [38]. This species tends to inhabit more open landscapes where it scavenges and hunts for food, and is often observed in human-modified habitats [56]. As such, a distinct overlap zone occurs across the Amazonian rainforest between approximately 0° through 7° latitude dividing the northern and southern subspecies [57]. However, contemporary taxonomic revisions based on molecular analyses no longer recognize these two distinct subspecies, due to low evidence of genetic differentiation, instead merging both of these into one single species, i.e., the 'Crested Caracara' (*C. plancus*) [57, 58]. Despite this, the species still persists in two distinct populations and it is likely that some individuals occasionally cross the overlap zone from either population. Largely, the species remains divided by the geographic overlap zone through the Amazonian rainforest, which may mediate differences in the ecology and diet of the two populations.

In an attempt to address differences in the diet of birds from both populations, we obtained the original subspecies-level range maps from the BirdLife International Datazone database (http://datazone.birdlife.org/home; previously updated in 2023) and plotted these using DIVA-GIS [59], where we assigned each photograph a population classification (either 'northern' or 'southern') based on geographic locations of our photographic data points. No updated

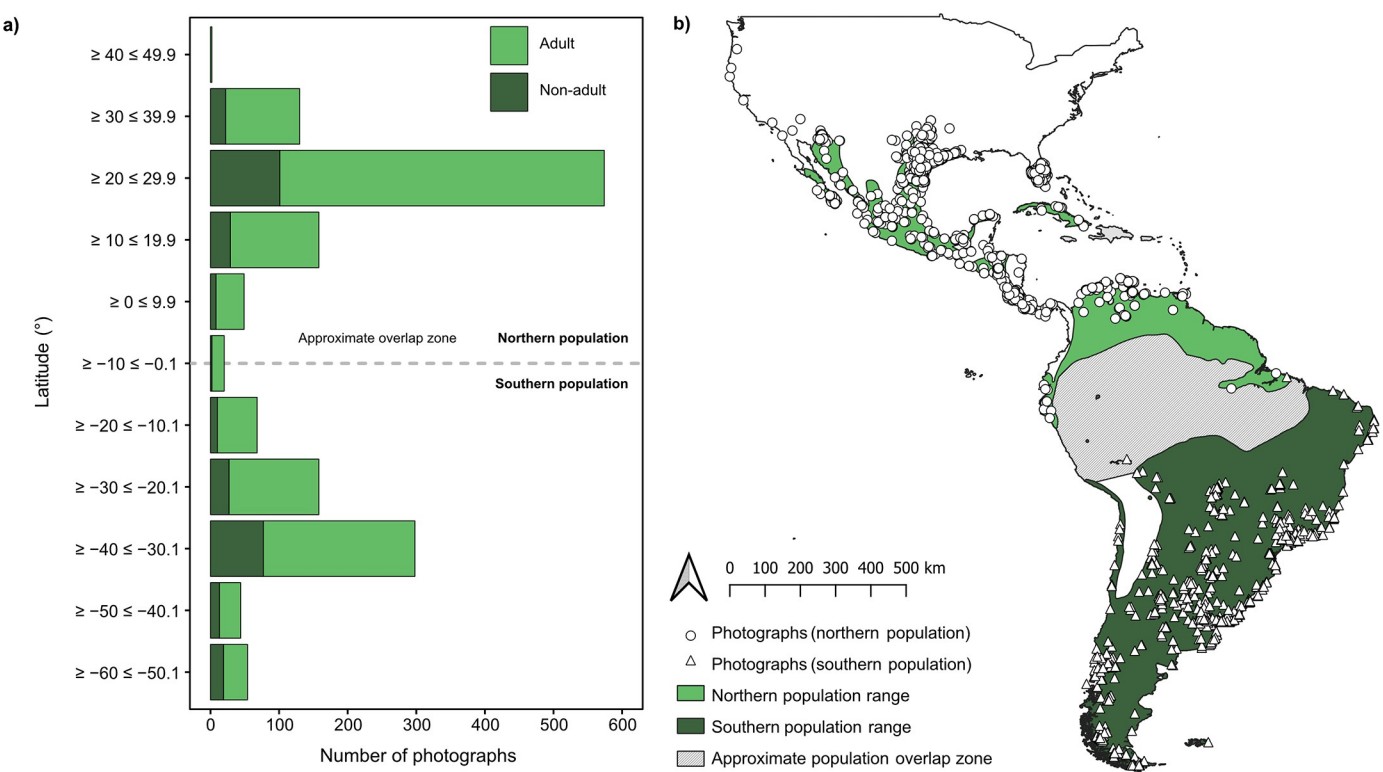

**Fig 1.** a) Latitudinal spatial distributions of 1,555 photographs of Crested Caracaras (*Caracara plancus*) feeding on identified food groups throughout North, Central and South America between 1987 through 2022. Photograph samples stacked by caracara age (Adult vs. Non-adult) and approximate location of the overlap zone occurring between approximately 0° through 7° latitude. b) Spatial distribution of photographs with an approximate depiction of the overlap zone occurring between 0° and 7° latitude between the two populations (Northern vs. Southern). Note—latitudinal categories presented in a) do not correspond to spatial data shown in b). Species ranges maps were provided by the BirdLife International Datazone [81].

range maps are available for the recent taxonomic revision of the species, therefore, we relied on the original subspecies range maps and interpreted these as proxies for the distinct distributions of the two continental populations. We plotted all useable photographs in QGIS version 3.14.16 [60] and conducted an overlap analysis between the geo-referenced photograph data points and each population range polygon (Fig 1a and 1b). Data points (N = 60) located within the approximate overlap zone between the northern and southern populations were excluded from the analyses as we could not accurately assign them a population.

## Spatial clustering

To account for potential pseudoreplication and non-independence between data points, i.e., photographs that are spatially distributed closely together may represent the diet of the same bird, we created spatial clusters. First, we computed 5,000 randomly distributed geographic data points across the entire Americas with a minimum spacing between points of 200 km. Caracaras tend to be a territorial species that display high site fidelity and occupy distinct home ranges of approximately 20,000 km$^2$ (*ca.* 100 km in radius) [32], hence we spaced each random point by at least 200 km to prevent overlap. We then calculated 100 km$^2$ radii buffer circles around each random point, which we interpreted as proxies of caracara home ranges. Next, we performed an intersection analysis in QGIS overlaying the caracara photograph data points with the home range proxies and assigned clusters of intersecting photographs a unique

cluster ID. This was performed under the assumption that each cluster ID likely represents photographic data points from the same bird.

## Statistical analyses

**Age and population effects on caracara diet.** All statistical analyses were conducted in R version 4.2.2 [61]. Data for food groups with $\geq 30$ observations were included, therefore, a total of 30 photographs were omitted due to small sample sizes (amphibians: N = 14; plants: N = 16). We ran a multinomial log-linear model using the R package "nnet" [62] to explore age and population-level associations on the probabilities of food groups in the photographic data set. We fitted 'food group' (birds, fishes, garbage, invertebrates, mammals or reptiles) as the response term and the interaction between 'age' (adult or non-adult) and 'population' (northern or southern) as the explanatory term (S1 Table).

**Latitudinal effects on caracara diet.** To explore latitudinal effects on the probabilities of different food groups within the caracara diet, we created multiple binary response variables for each food group. Each photograph was scored as either 1 or 0 depending on the presence or absence of a particular food group in the photograph. For example, photographs of a caracara feeding on a bird species were scored a '1' in a new binary variable termed 'birdbin' and those that did not include a bird were scored '0'. This was repeated for all food groups and for both population (northern or southern) subset data sets. Next, we ran a series of generalized linear mixed models (GLMMs) using the "lme4" package [63], with the binary food group variable fitted as the response terms, 'latitude' fitted as the explanatory terms and "cluster ID" fitted as random terms to account for potential non-independence between data points (S1 Table). These models explored the effects of latitude on the probability of each unique food group in the species' diet while accounting for potential pseudoreplication and non-independence between closely distributed data points. Models were run using binomial data distributions with 'logit' link functions (S1 Table). We created subset data sets for both northern and southern populations, and only ran the GLMM models for food groups with more than 30 observations. For the northern population, we ran models for the following food groups: 'birds', 'fishes', 'garbage' (defined as any non-natural human sourced food or waste), 'mammals' and 'reptiles'. For the southern population, we ran models for the following food groups: 'birds', 'fishes', 'invertebrates', 'mammals' and 'reptiles'. Due to a limited sample size for photographs of invertebrate food items from the northern population (N = 31), we were unable to assign individual cluster IDs for these photographs. Therefore, we ran a generalized linear model (GLM) to explore latitudinal effects on the probability of this food group within photographs of caracaras. Similarly to the GLMMs, the binary food group variable was fitted as the response term and 'latitude' fitted as the explanatory term. This single GLM was run with a binomial data distribution with a 'logit' link function (see S1 Table). We used the 'emmeans' package [64] to calculate the probabilities (± 95% confidence intervals (CI)) of different food groups within the caracaras' diet and to undertake additional *post hoc* contrasts (see S1 Table for an overview of the statistical modeling process).

## Results

### Overview of photographic samples

A total of 78,059 photographs of caracaras, taken across the Americas between 1987 through 2022, were downloaded from Macaulay Library (N = 42,488) and iNaturalist (N = 35,571) and were visually examined. Approximately 4.4% (N = 3,451) (Macaulay Library = 2,454; iNaturalist = 997) of the photographs appeared to include caracaras feeding and were subsequently marked as 'of interest'. Of these photographs, 68.9% (N = 2,381; Macaulay Library = 1,456;

**Table 1. Photographic sample sizes of Crested Caracaras (*Caracara plancus*) feeding throughout North, Central and South America by population and prey group sourced from Macaulay Library and iNaturalist between 1987 through 2022.**

| Food group | Population | | Total |
|---|---|---|---|
| | Northern | Southern | |
| | N (%/pop.) | N (%/pop.) | N (%/Total) |
| *Macaulay Library* | | | |
| unknown* | 295 (32.2) | 156 (28.8) | 451 (31) |
| mammals | 241 (26.3) | 140 (25.9) | 381 (26.2) |
| birds | 124 (13.6) | 101 (18.7) | 225 (15.5) |
| fishes | 89 (9.7) | 83 (15.3) | 172 (11.8) |
| reptiles | 79 (8.6) | 19 (3.5) | 98 (6.7) |
| invertebrates | 30 (3.3) | 24 (4.4) | 54 (3.7) |
| garbage | 38 (3.2) | 12 (2.2) | 50 (3.4) |
| amphibians* | 10 (1.1) | 3 (0.6) | 13 (0.9) |
| plants* | 9 (1) | 3 (0.6) | 12 (0.8) |
| Macaulay Library Total (%/ML Total) | 915 (62.8) | 541 (37.2) | 1456 (61.2) |
| *iNaturalist* | | | |
| unknown* | 208 (39.2) | 137 (34.7) | 345 (37.3) |
| mammals | 140 (26.4) | 91 (23) | 231 (25) |
| birds | 66 (12.5) | 78 (19.7) | 144 (15.6) |
| fishes | 54 (10.2) | 54 (13.7) | 108 (11.7) |
| reptiles | 37 (7) | 13 (3.3) | 50 (5.4) |
| invertebrates | 8 (1.5) | 6 (1.5) | 14 (1.5) |
| garbage | 14 (2.6) | 14 (3.5) | 28 (3) |
| amphibians* | 1 (0.2) | 0 (0) | 1 (0.1) |
| plants* | 2 (0.4) | 2 (0.5) | 4 (0.4) |
| iNaturalist Total (%/iNat. Total) | 530 (57.3) | 395 (42.7) | 925 (38.8) |
| Total (%/Total) | 1445 (60.7) | 936 (39.3) | 2381 (100) |

*Photographs from these food groups were omitted prior to statistical analyses due to small sample sizes or being unable to be assigned a broad food group.

iNaturalist = 925) were classified as 'useable' as it was confirmed that they contained a caracara feeding. Of all useable photographs, there were a far higher proportion of adults (N = 1,948; 81.8%) than non-adults (N = 433; 18.2%) and photographs were distributed across both population ranges (Table 1) with distributions of photographs for both populations peaking between approximately $\geq 20 \leq 29.9°$ (northern population) and $\geq -40 \leq -30.1°$ latitude (southern population) (Fig 1a and 1b). A total of 1,445 (60.7%) photographs were from the northern population and 936 (39.3%) from the southern population (Table 1). For the northern population 1,214 (84% northern population) photographs were of adults and 231 (16%) included non-adults (see Fig 2a–2f for examples). For the southern population, 734 (78.4% southern population) photographs were of adults and 202 (21.6%) included non-adults. There were 796 photographs that we were unable to assign to a broad food group, and together with the 30 photographs representing amphibian and plant food groups, these were excluded prior to the analyses. We were able to assign broad food groups to 1,555 (65.3% of useable photos) photographs which were used in the analyses (Table 1).

## Dietary composition

Overall, caracaras were photographed feeding on most major taxonomic groups across the range of both populations (Table 1). Of the 1,555 photographs containing identified food

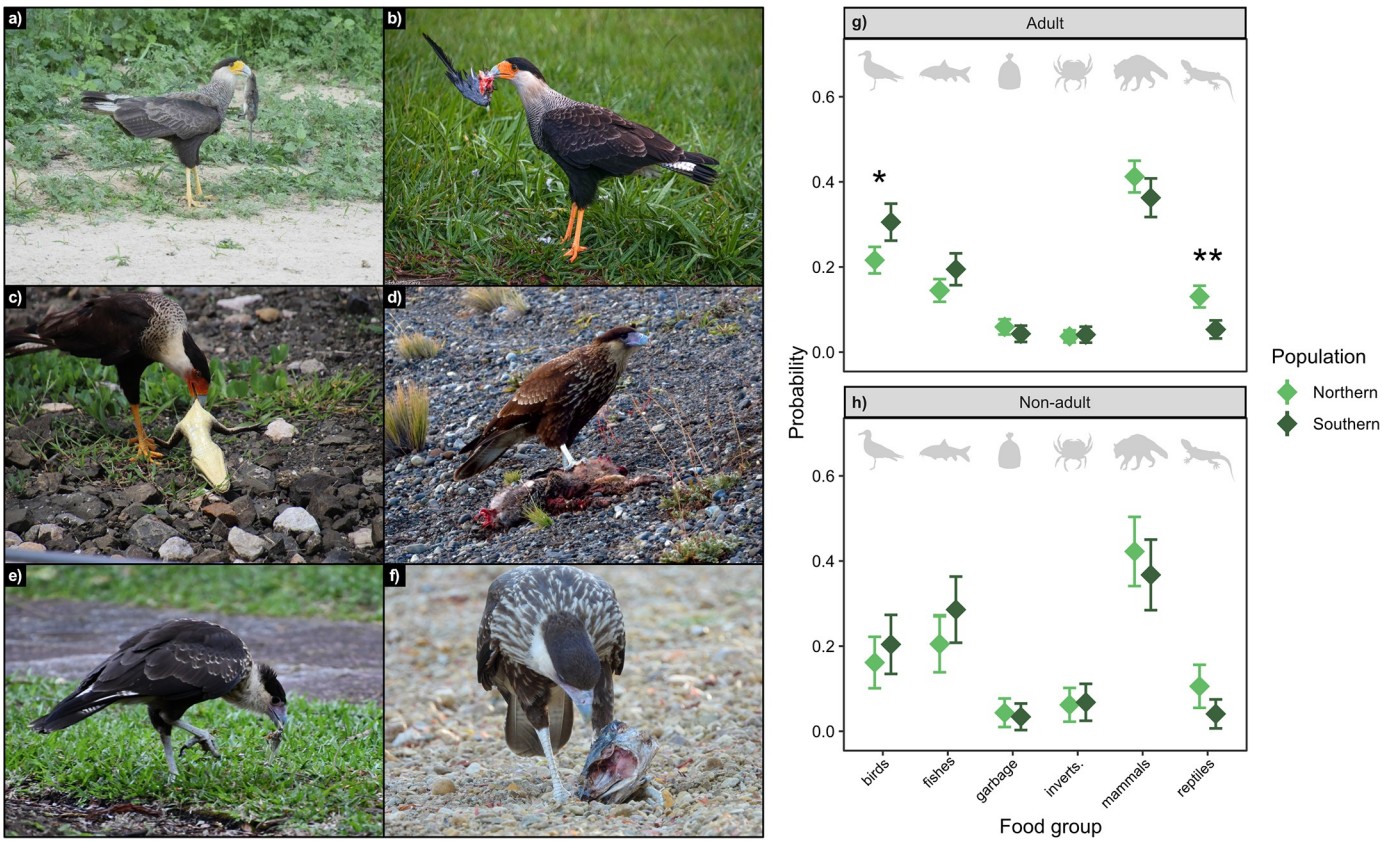

**Fig 2. Age differences present in 1,555 photographs of Crested Caracaras (*Caracara plancus*) feeding throughout North, Central and South America between 1987 through 2022.** a) Adult caracara feeding on a mammal (photograph credit: lucianomassa CC BY NC-SA iNaturalist), b) adult feeding on bird remains (eduardopaiva CC BY-BC-SA iNaturalist), c) adult feeding on crocodilian remains (isisanimals CC BY-NA-SA iNaturalist), d) non-adult feeding on a mammalian carcass (ericgalas CC BY-NA-SA iNaturalist), e) non-adult feeding on an invertebrate (varvarenja CC BY-NA-SA iNaturalist) and f) non-adult feeding on a fish carcass (mariocastanedasanchez CC BY-NA-SA iNaturalist). Panels g) and h) show age and population differences in the probabilities of each food group within photographs of caracaras feeding. Significant differences indicated with asterisks: '*' $P < 0.05$ and '**' $P < 0.01$. Error bars represent 95% confidence intervals.

groups, mammals were recorded in 612 (39.4%) photographs followed by birds (369; 23.7%), fishes (280; 18%), reptiles (148; 9.5%), garbage (78; 5%) and invertebrates (Table 1).

## Age and population effects on caracara diet

There was no significant age effect (age: $X^2_{1,5} = 8.071$, $P = 0.152$) or a significant interaction effect (age × population: $X^2_{1,5} = 0.172$, $P = 0.999$) between caracara age and population on the probabilities of different food groups in the photographic data set (S2 Table). However, there was a significant population effect (population: $X^2_{1,5} = 36.431$, $P < 0.0001$) with increased probabilities of reptiles being included in photographs taken of adult caracaras from the northern population relative to adult birds from the southern population ($t_{1,20} = 4.334$, $P = 0.002$) but the opposite pattern for birds ($t_{1,20} = -3.48$, $P = 0.012$) (Fig 2g and 2h; S2 Table).

## Latitudinal effects on caracara diet

For photographs taken of individuals from the northern population, there was a negative latitudinal effect with the probability of fishes ($z_{1,533} = -2.251$, $P = 0.024$) and invertebrates ($z_{1,918} = -4.616$, $P = < 0.0001$; S4 Table) within the diet increasing towards to the equator (Fig 3a and

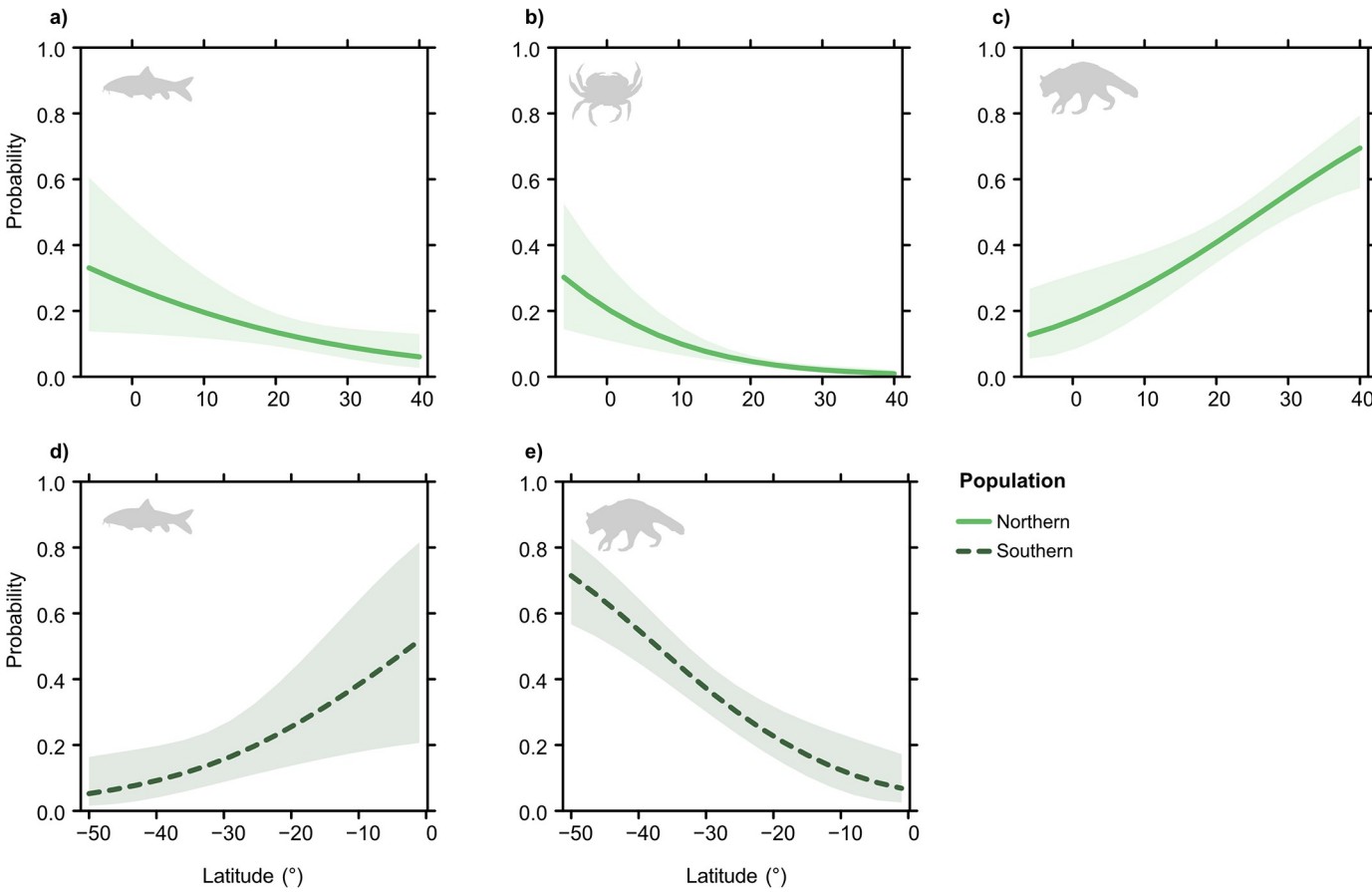

**Fig 3.** Significant predicted latitudinal effects, with 95% confidence intervals, on the probability of a) fishes, b) invertebrates and c) mammals within photographs of Crested Caracaras (*Caracara plancus*) from the northern population feeding between 1987 through 2022. For the southern population, significant latitudinal effects for d) fishes and e) mammals within photographs of caracaras also shown.

3b; S3 Table). Contrastingly, the opposite pattern was observed for mammals with an increased probability away from the equator towards higher latitudes ($z_{1,533} = 3.974$, $P < 0.0001$) (Fig 3c; S3 Table). There were no significant latitudinal effects on the probability of birds, garbage or reptiles in the diet of caracaras from the northern population (S3 Table).

Similarly, for photographs taken of individuals from the southern population, the probability of fishes being included in photographs increased towards the equator ($z_{1,376} = 2.467$, $P = 0.014$) (Fig 3d; S3 Table). Furthermore, there was an increased probability of mammals being included in photographs of individuals from the southern population with increasing distance away from the equator ($z_{1,376} = -4.432$, $P < 0.0001$) (Fig 3e; S3 Table). There were no significant latitudinal effects on the probabilities of birds, invertebrates or reptiles in photographs of caracaras from the southern population feeding (S3 Table).

## Discussion

### Crested caracara diet varied along latitudinal gradients

Our study represents the first continental-scale quantitative diet assessment for caracaras, throughout the species' range across the Americas. As expected, for both caracara populations the probability of mammals in photographs increased with increasing distance away from the

equator. There was a significant increase in the probability of reptiles in the diet of caracaras from the northern population, and the opposite pattern in the probability of birds in the diet of the southern caracara population. Although we suspect this may be a confounding effect of both larger sample sizes per-unit-area when compared to the southern population and availability of suitable habitat for reptiles especially across Mexico [65]. Contrary to patterns relating to global climatic and species diversity patterns along latitudinal gradients [50–52], the probability of reptiles showed no significant increase towards the equator for either population. However, there were significant latitudinal effects in the probabilities of fishes and invertebrates in photographs of northern caracaras, increasing towards the equator. These patterns may be driven by increased species richness within these groups towards the equator, and thus a presumed increase in availability in tropical/subtropical environments relative to temperate zones [66]. However, more research is needed to decipher these patterns.

## Comparisons between other diet studies

Compared with previous studies, our findings confirm that caracaras are a generalist species that feed on most major taxonomic groups [39–48, 67]. Analysis of 299 prey remains found that mammals (31.4%), reptiles (24.1%), fish (23.7%), birds (13.4%) and amphibians (7.4%) represented the most numerous groups within the species' diet in south-central Florida [40], showing some consistency with our findings from across the species' range. Unlike another study of caracara diets in Andean Patagonia which concluded that adults tended to take smaller prey (such as arthropods) while supplying juvenile birds with larger vertebrate prey, there were no dietary differences between age classes in our data set [38]. This may be explained by the lack of observations of adults feeding young or that our approach did not allow us to distinguish between adult birds feeding themselves as opposed to taking food items to the nest. Other studies report that insects [41, 42, 44] and plant material (such as Pecans (*Carya illinoinensis*) [43] and palm fruits [68]) also featured within the species' diet, however, fewer than 3% and 1% of food items in our study were attributed to these food groups, respectively.

## Web-sourced photography is a useful tool to study raptor diets across large spatial scales

Apart from a previous study on Martial Eagles [27], which also used web-sourced photography, relatively few studies have explored raptor diets at large geographic scales. Other studies have attempted to do so by compiling dietary data from multiple separate studies across a species' range, providing useful comparisons especially for our data on latitudinal trends [49, 69]. For example, a global examination of Western Barn Owl (*Tyto alba*) and American Barn Owl (*T. furcata*) diets found a positive relationship in the proportion of mammal prey in colder environments [25]. Unlike previous studies which studied the diets of Eurasian Sparrowhawk and Martial Eagles [27, 29], we did not find any age effects in the diet of caracaras which may be explained by their generalist nature and tendency to feed on carrion [47, 67]. Eurasian Sparrowhawk are avian specialists [27, 28] and age-related differences in diet may be more pronounced. Similar to our findings, a continental assessment of Montagu's Harrier (*Circus pygargus*) diet found that mammalian prey increased at higher latitudes due to the prevalence of agricultural land cover at more northern latitudes [49]. A separate study on Montagu's Harriers [70] combined the use of web-sourced photography with pellet analyses to explore seasonal-, regional- and sex-differences in the species' winter diet. The proportion of key food groups in the species' diet was largely consistent using both methods [70], demonstrating that

using web-sourced photography alongside more traditional methods can improve studies of raptor diet.

## Study limitations

Our approach, like most methods to study raptor diets [8, 71], has been shown to be potentially biased towards larger and more identifiable food items, e.g., mammals and birds, which require longer processing times [28, 29]. Consequently, this food-size bias may partly explain why we detected low relative proportions of smaller prey groups, such as invertebrates (2.9%), plant material (0.7%) and amphibians (0.6%) within our photographs. Furthermore, our approach assumes that the food items included in the photographs were actually consumed by caracaras. Due the lack of ability to ascribe individual cluster IDs to photographs of northern caracaras feeding on invertebrates, we were unable to account for potential pseudoreplication with a potential risk of non-independence between invertebrate data points. Spatial coverage of our photographs spanned the geographic ranges of both caracara populations. However, it is likely that a confounding variable and a sampling bias towards more urbanized habitats, roadsides or areas with higher human activities may persist in our data set. These locations are where more people may tend to take photographs, and food type and availability may potentially be affected by the degree of urbanization. A key assumption when using web-sourced photography pertains to the species being well-sampled. This approach, unlike more traditional techniques, e.g., analysis of pellet remains, is unsuitable when studying cryptic species or those that are challenging to observe in the field. Therefore, strengths lie in its ability to be used in conjunction with other methods complementing future raptor research [70]. Potential opportunities for future research include studying differences in urban and non-urban raptor diets using our approach. As with most methods that study the diets of predatory species that also scavenge, it is challenging to distinguish between food items that were hunted by caracaras and those which were scavenged. Furthermore, due to incompatibilities with small and uneven sample sizes, we were unable to account for differences in food groups within photographs of caracaras from smaller subpopulations, such as on the Falkland Islands [72].

## Utilizing citizen science data can be a time- and cost-effective method to study raptor diets

Unlike more traditional methods to study raptor diet, web-sourced photography provides novel perspectives when quantifying raptor diets across large spatial scales. Traditional methods are often constrained to a few breeding pairs during the breeding season and unlikely represent dietary patterns for the entire population which includes floaters, sexually immature birds and individuals that fail to breed. Such approaches do not allow for the exploration of diet outside of the breeding season, which has been shown to change in response to prey availability [73] and changes in behaviour of adult birds [28]. Our approach is a time- and cost-effective method to study raptor diets, requiring minimal skill from the observer other than knowledge of key food items and access to the internet [23]. As a result, the increasing prevalence of open-source intelligence data in daily life [74], provides an accessible approach for researchers without access to large funds, time commitments required or the expertise necessary for long-term monitoring of multiple populations. Assuming that the study species is well-sampled this approach can be applied to many other predatory species, including those that are inconspicuous or more challenging to study in the field, such as snakes [75], providing novel perspectives on our understanding of predator ecology.

## Conclusions

The findings of our study represent the first assessment of caracara diet throughout the species' geographic range across the Americas. We were able to explore dietary differences between caracara age classes and between populations, which has not been attempted before for this species. Across their range, caracaras fed from most major taxonomic groups, and we found significant latitudinal effects on the probabilities of fishes, invertebrates and mammals within photographs of caracaras feeding which likely reflect latitudinal variations in climate and land cover type. Web-sourced photography complements other methods to study raptor diets, such as DNA metabarcoding [4, 19, 20] or stable isotope analysis [16–18], maximizing sampling effort across taxonomic groups that may be difficult to identify from a single photograph, e.g., insects. The role of open-source intelligence and citizen science data has expanded in ornithology and continues to increase in popularity due to the relative ease of accessing data, large sample sizes and increasing quality of available metadata. Continued availability of open-source intelligence and citizen science data provides a wealth of information on predator diets [27–29, 70, 76], interspecific interactions [30], polymorphism [77, 78], conservation policy and planning [79], species range shifts [80], and has the potential to benefit ongoing and future raptor research across large spatial-, demographic- and temporal-scales.

## Supporting information

**S1 Table. An overview of the statistical modeling process undertaken to explore the effects of age, population and latitude on the diet of Crested Caracaras (*Caracara plancus*) throughout North, Central and South America between 1987 through 2022.** 'GLM' = Generalized Linear Model, 'GLMM' = Generalized Linear Mixed Model.
(DOCX)

**S2 Table. Contrasts from the multinomial log-linear model exploring the effects age, population and their interaction (age × population) on the probability of different food groups in photographs of Crested Caracaras (*Caracara plancus*) feeding throughout North, Central and South America between 1987 through 2022.** Significant contrasts ($P < 0.05$) highlighted in **bold**.
(DOCX)

**S3 Table. Outputs from the generalized linear mixed models exploring the effect of latitude on the probabilities of different food groups within photographs of Crested Caracaras (*Caracara plancus*) feeding throughout North, Central and North America.** SE = standard error, df = degrees of freedom. Significant effects ($P < 0.05$) in **bold**.
(DOCX)

**S4 Table. Outputs from the generalized linear model exploring the effect of latitude on the probability of invertebrates within photographs of Crested Caracaras (*Caracara plancus*), from the northern population only, feeding throughout North, Central and North America.** SE = standard error, df = degrees of freedom. Significant effect ($P < 0.05$) in **bold**.
(DOCX)

## Acknowledgments

The authors are thankful to all photographers who facilitate research such as this by uploading photographs of caracaras onto open access databases such as Macaulay Library and iNaturalist. In addition, we would like to thank T. Katzner, J. Morrison, E. Miranda, D. Tubelis and the anonymous reviewers for their helpful comments on earlier drafts.

## Author Contributions

**Conceptualization:** Connor T. Panter, Vincent N. Naude.

**Data curation:** Connor T. Panter, Vincent N. Naude.

**Formal analysis:** Connor T. Panter.

**Investigation:** Connor T. Panter, Vincent N. Naude, Facundo Barbar.

**Methodology:** Connor T. Panter, Vincent N. Naude, Arjun Amar.

**Project administration:** Connor T. Panter, Vincent N. Naude.

**Software:** Connor T. Panter, Vincent N. Naude.

**Supervision:** Arjun Amar.

**Validation:** Connor T. Panter, Vincent N. Naude, Facundo Barbar.

**Visualization:** Connor T. Panter, Vincent N. Naude.

**Writing – original draft:** Connor T. Panter, Vincent N. Naude.

**Writing – review & editing:** Facundo Barbar, Arjun Amar.

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
