## [Decision Letter · Decision Letter 0]

19 Jan 2024

PONE-D-23-42552Using web-sourced photographs to explore continental-scale dietary patterns in a New World raptor, the Crested Caracara (Caracara plancus)PLOS ONE

Dear Dr. Connor Panter,

Thank you for submitting your manuscript to PLOS ONE. After careful consideration, we feel that it has merit but does not fully meet PLOS ONE’s publication criteria as it currently stands. Therefore, we invite you to submit a revised version of the manuscript that addresses the points raised during the review process. Thank you for submitting to PLOS ONE your study on dietary patterns of a raptor species, based on community science data. We received three reviews, and I also reviewed it as I had published a few papers by using a similar approach.

Reviewer 1 has suggested Minor Revision. Numerous suggestions have been provided to improve the quality of your manuscript. His/her main concerns are: 1) the restriction of your sample to the Macaulay Library database (as there are others for the Neotropical region); 2) the absence of examination of seasonal variation in the occurrence of food items in the diet of the raptor species.

Reviewer 2 has suggested Minor Revision and provided numerous compliments on your research, and a wide range of suggestions to improve the quality of the manuscript. Note that he/she is surprised with the fact that you overlooked the WikiAves and iNaturalist databases/platforms.

Reviewer 3 has suggested Rejection mainly based on the fact that you gave substantial emphasis on the methodology, what he/she considers that makes your (interesting) study more suitable for an ornithological journal, and not for PLOS ONE. There are some suggestions that you can consider. I think that a rejection is not necessary, but you will need to make some changes in the Introduction and Discussion (reduce the emphasis on the methodology and increase the number of examples/studies and comparisons that examined latitudinal variation in the diet of birds).

I suggest that you consider these four reviews (please find mine below) to verify with what you agree, and thus follow. There is a great potential for the publication of a high quality study in PLOS ONE.

Dárius P. Tubelis

PLOS ONE Editor Please submit your revised manuscript by 05 March 2024. If you will need more time than this to complete your revisions, please reply to this message or contact the journal office at plosone@plos.org. Please include the following items when submitting your revised manuscript:A rebuttal letter that responds to each point raised by the academic editor and reviewer(s). You should upload this letter as a separate file labeled 'Response to Reviewers'.A marked-up copy of your manuscript that highlights changes made to the original version. You should upload this as a separate file labeled 'Revised Manuscript with Track Changes'.An unmarked version of your revised paper without tracked changes. You should upload this as a separate file labeled 'Manuscript'.

We look forward to receiving your revised manuscript.

Kind regards,

Dárius Pukenis Tubelis, Ph.D.

Academic Editor

PLOS ONE

2. In your Methods section, please include additional information about your dataset and ensure that you have included a statement specifying whether the collection and analysis method complied with the terms and conditions for the source of the data.

4. Please remove your figures from within your manuscript file, leaving only the individual TIFF/EPS image files, uploaded separately. These will be automatically included in the reviewers’ PDF.

5. We note that Figure 2 in your submission contain [map/satellite] images which may be copyrighted. All PLOS content is published under the Creative Commons Attribution License (CC BY 4.0), which means that the manuscript, images, and Supporting Information files will be freely available online, and any third party is permitted to access, download, copy, distribute, and use these materials in any way, even commercially, with proper attribution. For these reasons, we cannot publish previously copyrighted maps or satellite images created using proprietary data, such as Google software (Google Maps, Street View, and Earth). For more information, see our copyright guidelines: http://journals.plos.org/plosone/s/licenses-and-copyright.

Additional review by Dárius Tubelis:

Your study is very interesting and highlights that citizen science data (photographs) can be used to investigate dietary aspects of birds at large spatial and temporal scales in the Neotropics. The approach is quite appropriate and the results are interesting.

I´m mainly concerned with the restriction of your sample to data from the Macaulay Library (that encompasses eBird). For example, the WikiAves platform (https://www.wikiaves.com.br) harbors a lot of data about Brazilian birds, including 27,100 records of Caracara plancus. On 08 January 2024, I clicked on “Registros” (Records), and then on “Busca Avançada” (Advanced search). Then, I typed the name of the species (“Espécie”, Species) and then, near the page bottom, clicked on the option “Alimentando-se/caçando” (Feeding/hunting). This search resulted in 3,429 photographs of C. plancus. Of these, about 50% might have evidence of feeding activities by this raptor in Brazil (I guess this percentage based on studies that I have done using this platform – e.g., Rupornis magnirostris). This available amount of records would represent a good sample of food items along a considerable latitudinal range, in your study. Another option with less numerous records for South America would be iNaturalist….these two additional searches could double your sample size, or do even more….please think about (R1 and R2 also pointed out this question). The current sample could lead to negative criticism by readers, despite the large number of records/photographs. On other hand, your results appear to be quite robust (convincing), and thus your Macaulay-based sample would be enough. Please try to del with this aspect in the Methods and Discussion sections.

Abstract.

Ok.

Introduction.

Very well structured with relevant publications. But note that you provided few sentences regarding latitudinal variation in the diet of birds, including raptors. For example, along lines 70-80 you can briefly present the major results of these investigations at large spatial scales. After the current examples, you could add 2-3 examples mainly focusing latitudinal variation. Also, try to reduce the amount of information on the methods relative to diet of birds. The reading is running well, just need to do these modifications for a better suitability to PLOS ONE.

Line 51. Is there a word better than “however” ?

Line 90. “it how diet” is correct ?

Objectives. Ok.

Material and Methods

A Study Area section would be welcome. Please try to explain main aspects of the Americas that are pertinent to your study, such as North, Central, and South Americas, maximum values of Lat in both Hemispheres, the line of Equator position. It would be useful for readers from other parts of the world.

Section “Web-sourced…”.

If you decide to keep only with the Macaulay Library data, you will have to explain why you did not included searches in the WikiAves, iNaturalist and other databases. Like tell that you sample is large, huge, when compared with those of previous studies. On the other hand, if you decide to include other databases, then add text to explain how you did the searches.

Line 116. You repeated…why not use “For them….” ?

Line 117. What did you consider (adult or non-adult) when you could see an adult with a prey near a nestling in the nest ? This type of photograph has your two types of caracaras….

Line 188. I did not understand item 4 (location)…can you briefly explain within brackets ?

Line 123. These brackets are too distant. Do you really need them ? Six lines within them…can you try to reduce this text (Lines 121-130)?

Section “Prey and age”.

Line 141, if at least two authors, than should be more than CP and FB…please check the writing.

Line 142. Better if you start a new paragraph with “Adult caracaras…”.

Next lines. It is not clear if you included nestlings or not. It appears that you included only juveniles outside nests, and adults. But some photographs might show 1-2 adults in nests with prey and nestlings, as I could note in WikiAves. Please clarify it.

Section “Population”.

Lines 152 and 154. Extended or extends?

You could divide this section in 2-3 paragraphs. It is too long.

Calling a figure with a map would be welcome here.

Section “statistical Analysis”

How did you control for potential influence of seasonality on these results ? For example, you might have more photographs in the rainy or dry season for a given caracara population… please note that Reviewer 1 has concern about this, or similar potential influence on data.

Line 238. You have to place Table 1 here.

Results.

Line 244. As you write “Of these…”, this percentage would be 1501/2454). No ?

Lines 250. You have to call “(Fig 1)”, abbreviated with no dot.

In the end of this paragraph (Line 258), you have to bring the captions of Fig 1 and Fig 2 (but not the figures, that should be kept in the end of the manuscript). Please check Instructions again.

Line 277. Bring Table 2 here.

Line 287. Bring here: Caption of figure 3…

Discussion

It would be easier to read if you use subtitles for sections of the Discussion.

Make sure that you discuss more extensively with previous study on the diet of the species, influence or not of age among raptors, influence of latitude on raptor diets. Your current discussion is a bit poor relative to these topics. Try to the emphasis on methodology.

Always consider an international readership by citing studies conducted in several ecoregions, countries, involving several species.

Both reviewers made some comments that might be useful for you here in the Discussion. For example, R2 commented that this study can be considered an example for the study less common, or more “difficult” species.

References.

The correct is to use like this “2. Smith TC….” Instead, you used “[2] Smith TC….”.

Page numbers. There is no space before them (after the volume and :).

Page number. You have to use a long dash between then, not an hifen (-).

DOIs. You have to provide it for all references that have it. Follow the format: (http://doi.org/10.....).

Ref 6. The initials in capitals should not occur, except for the first word and names….check for all.

Ref 26. It occurs with a duplicate. You will need to change all these numbers here and in the text.

Please also check Instructions for books, chapters, websites….

Tables.

They should be moved to the text, by the end of the paragraph that cite them.

Figures.

Their captions should be in the text, as above for tables. Keep figures here in the end with their “Fig x”.

Fig 1. Can you increase a bit the size of letters and numbers ? Maybe, titles of axes in bold. Is “divide” correct ? (along the dashed line).

Fig 2. You used a duck, but do caracaras prey on them ? What about a pigeon or other smaller bird ?

Reviewers' comments:

Reviewer's Responses to Questions

**Comments to the Author**

1. Is the manuscript technically sound, and do the data support the conclusions?

Reviewer #1: Yes

Reviewer #2: Yes

Reviewer #3: Yes

2. Has the statistical analysis been performed appropriately and rigorously? 

Reviewer #1: Yes

Reviewer #2: Yes

Reviewer #3: Yes

3. Have the authors made all data underlying the findings in their manuscript fully available?

Reviewer #1: Yes

Reviewer #2: Yes

Reviewer #3: Yes

4. Is the manuscript presented in an intelligible fashion and written in standard English?

Reviewer #1: Yes

Reviewer #2: Yes

Reviewer #3: Yes

5. Review Comments to the Author

Reviewer #1: I thank the editor for inviting me to review this manuscript. The authors have presented the background, methods, results and discussion in a technically sound manner. I have a few suggestions for some of the sections which I believe will help add more clarity to the manuscript. The detailed comments are as follows:

Introduction

Line 44-45: The sentence can be rephrased – “Food availability and abundance can affect population dynamics of raptors”

Line 50: “provisioning” instead of “provision”

Line 57-59: It will be good to add some citation. Suggestion: https://link.springer.com/article/10.1007/s10344-010-0480-z

Line 77-79: This sentence can be deleted. Information repeating in the beginning of the para.

Line 90: “it” can be removed from the last sentence

Line 91: Will be good to add information on status of the species in the Americas, especially since it occupies open habitats which are under threat globally.

Methods

Line 111: Any justification as to why only Macaulay library was considered for collecting photographs? What about other social media databases such as Facebook, Flickr etc.?

Line 169: Which year is the latest update of this range map?

Line 184: Were these data points location data? It will be good to specify for clarity.

Results

Line 261-264: Can this information also be presented in Table 1 separately? The table currently has information on 1501 photographs. But food groups could be assigned to only 1029 right?

Line 274: This result has not been discussed in detail in the discussion section. What might be the reason for higher probability of reptiles in the northern population diet?

Discussion

Line 303-306: Will it be good to test for seasonality of occurrence of these food groups from photographs? Maybe occurrence of invertebrates in diet might be more during the warmer months in areas away from the equator. The availability of these food sources might be limited to warmer seasons in the temperate regions.

Lines 315-318: This reference is missing in the literature cited. I believe it’s this study: https://doi.org/10.1006/jare.2000.0745

Line 329: There are two citations with the same number [26]. This has to be rectified in the references section.

Figures

Figure 1: Along with the latitudinal degrees, it will also be good to show the direction on the x-axis (N/S).

Figure 2 a.: The lat longs can be shown in the form of a grid on the map along with the equator line. Since the major part of the study is about latitudinal variation and possible differences in diet in regions away from the equator.

Figure 2 b.: The “**” is not clearly visible on the graph. The font can be made larger.

Reviewer #2: The present study utilizes an online dataset as a tool to acquire knowledge about the caracara diet. The researchers employed robust statistics and large sample sizes to test various hypotheses regarding caracara foraging. They exhibited wit and precision in their test choices and interpretations. Based on their observations, the authors infer that the caracara diet remains consistent with existing descriptions; the species is a known generalist. However, the proof of concept they present allows for the application of the same methods to eagles or other predators for which dietary data might be both crucial and scarce—a point that warrants discussion. Additionally, other observed diet patterns strongly align with their associated data. This study, both simple and robust, epitomizes citizen-based science in my view. I believe PLOS ONE is a suitable publication venue for this manuscript and I wholeheartedly recommend it for publication. Congratulations.

L52: Please rephrase this section since it's not about a misinterpretation but rather a misunderstanding of the natural world.

L66-77: This trend extends to other predators like giant snakes and carnivores, among others.

L77-80: In essence, this provides an opportunity to study a species that have been extensively researched; which is no opportunity at all. Please note that the methods mentioned and criticized in the beginning of the previous paragraph aren't limited to easy species. It feels like cherry picking from your side in favour of your collection method; an effort to address this would enhance the text. Moreover, it makes the argument over the selection of your study species somewhat circular. None of this, in my opinion, is detrimental to the paper, but the text should be revised—especially concerning the choice of the study species.

L94: It surprises me that you chose to overlook larger databases such as WikiAves or iNaturalist. Could you provide some background on your choice?

L108-237: I'm impressed by the clarity, elegance, and robustness of the methods and sample design. The threshold sample sizes for dietary analysis were particularly impressive. However, the categorization of prey and other food sources feels somewhat artificial to me. I believe including functional groups—like domestic animals or aquatic species? L225-228—might make more sense. Nevertheless, congratulations on the excellent work.

L257: I'm absolutely impressed by the broad and extensive sample. I'd like the study to explore a rarer, threatened, or less-known species. However, this doesn't diminish the outstanding work you've done.

Reviewer #3: In the manuscript entitled: “Using web-sourced photographs to explore continental-scale dietary patterns in a New World raptor, the Crested Caracara (Caracara plancus)” the authors used web-sourced photography and explored continental-scale demographic and latitudinal dietary patterns between adult and non-adult Crested Caracaras throughout the species’ range across the Americas. The paper is well written, well structured and has valuable information given that on the one hand, there is little work of this type and even less in the southern hemisphere.

However, the manuscript, as it is written now, I consider it to be methodological in nature as the focus is on the advantages of web-sourced photographs, so it may not be of interest to a wider audience such as PlosOne.

In addition, I have some comments that may help to improve the manuscript:

Abstract-Introduction:

In the introduction, predictions are made as to what they expected to find in terms of age, latitude, mammal consumption and taxonomic groups, however this is not indicated in the abstract.

Line 75. Indicates that previous work examined the diet, using this approach?

Materials and Methods:

Línea 204-206. This information is redundant, this was already stated in the sentence above (with other words), I think it is better to remove it.

Results:

Line 276. Figure?

Discussion:

At the beginning of the introduction, they should have made reference to what they had indicated at the end of the introduction (line 98-line 101). They do so in line 300-3001.

In general, the discussion is focused on the methodology, therefore, although it is correct, I consider that it would not be of interest to a wide audience and an Ornithological journal would be more appropriate for this interesting work.

6. PLOS authors have the option to publish the peer review history of their article (what does this mean?). If published, this will include your full peer review and any attached files.

Reviewer #1: No

Reviewer #2: **Yes: **Everton Miranda

Reviewer #3: No

---

## [Author Response · Author response to Decision Letter 0]

25 Apr 2024

Response to reviewers PONE-D-23-42552

Comments from editor:

Your study is very interesting and highlights that citizen science data (photographs) can be used to investigate dietary aspects of birds at large spatial and temporal scales in the Neotropics. The approach is quite appropriate and the results are interesting.

I´m mainly concerned with the restriction of your sample to data from the Macaulay Library (that encompasses eBird). For example, the WikiAves platform (https://www.wikiaves.com.br) harbors a lot of data about Brazilian birds, including 27,100 records of Caracara plancus. On 08 January 2024, I clicked on “Registros” (Records), and then on “Busca Avançada” (Advanced search). Then, I typed the name of the species (“Espécie”, Species) and then, near the page bottom, clicked on the option “Alimentando-se/caçando” (Feeding/hunting). This search resulted in 3,429 photographs of C. plancus. Of these, about 50% might have evidence of feeding activities by this raptor in Brazil (I guess this percentage based on studies that I have done using this platform – e.g., Rupornis magnirostris). This available amount of records would represent a good sample of food items along a considerable latitudinal range, in your study. Another option with less numerous records for South America would be iNaturalist….these two additional searches could double your sample size, or do even more….please think about (R1 and R2 also pointed out this question). The current sample could lead to negative criticism by readers, despite the large number of records/photographs. On other hand, your results appear to be quite robust (convincing), and thus your Macaulay-based sample would be enough. Please try to del with this aspect in the Methods and Discussion sections.

>>> Thank you for taking the time to consider our manuscript, and for your positive and constructive review. During the revision process, we explored whether including additional photographs from WikiAves would be suitable for our study. We decided that due to the photographs being restricted geographically to Brazil, that this may introduce a spatial bias within our continental scale dietary assessment. Despite this, we did indeed increase our sample size by collating and processing photographs of caracaras from iNaturalist. Overall, we processed an additional 35,571 photographs from iNaturalist and together with our sample from Macaulay increased our initial sample size to 78,059. From the new photographs, after screening and removing duplicates, we added some 925 new photos to our analyses, resulting in a data set comprising 1,555 photographs used in the statistical analyses. Interestingly, our results remained largely the same when including the additional photographs. We increased our sample size for caracaras from the southern population substantially which enabled us to explore latitudinal effects in food groups that before had insufficient sample sizes. We hope that the editor agrees with us that our manuscript is much more robust compared to the initial submission.

Title – we have revised the title to shorten it and to focus on our ability to assess diet at the continental scale using our approach.

“Continental scale dietary patterns in a New World raptor using web-sourced photographs” (L1-2).

Abstract.

Ok.

>>> the abstract has been updated given the new results from the inclusion of iNaturalist data (L26; L30).

Introduction.

Very well structured with relevant publications. But note that you provided few sentences regarding latitudinal variation in the diet of birds, including raptors. For example, along lines 70-80 you can briefly present the major results of these investigations at large spatial scales. After the current examples, you could add 2-3 examples mainly focusing latitudinal variation. Also, try to reduce the amount of information on the methods relative to diet of birds. The reading is running well, just need to do these modifications for a better suitability to PLOS ONE.

>>> Thank you for these suggestions, we have expanded on latitudinal variation within bird and raptor diets in the suggested section while trying to also introduce the use of web-sourced photography as a tool to study diet. Upon reflection, we felt that this was appropriate at this point in the manuscript and provides a link to the following paragraph:

“For species that inhabitant large geographical areas, exploring how diet varies latitudinally may improve understanding of its ecology. For example, ecological and climatic conditions along latitudinal gradients influence the availability and presence of prey species within the wider environment [24], affecting the diversity and dietary composition for many species including birds [25]. Reviews of the raptor diet literature found distinct latitudinal patterns in the proportions of prey groups in some raptor diets, with increased probabilities of mammalian prey at higher latitudes [26,25].” (L68-75).

Line 51. Is there a word better than “however” ?

>>> To better emphasise the point that dietary assessments are dependent on the method and scale used, we have replaced the word “however” with “Most importantly” (L50).

Line 90. “it how diet” is correct ?

>>> Thank you for spotting this, we have now removed “it” from the sentence.

Objectives. Ok.

>>> No action required.

Material and Methods

A Study Area section would be welcome. Please try to explain main aspects of the Americas that are pertinent to your study, such as North, Central, and South Americas, maximum values of Lat in both Hemispheres, the line of Equator position. It would be useful for readers from other parts of the world.

>>> We have now revised the Methods accordingly and included a “Study Areas” subsection outlining the geographic extent of our study areas, regions and biogeographic subrealms: 

“Study Areas

Our study area encompasses the caracara’s entire geographic range, spanning a latitudinal gradient from approximately 45°N to -55°S, and a longitudinal gradient from approximately -35°E to -125°W. Photographic samples span multiple regions including the Nearctic of North America, Mexican Transition Zone through North/Central America and into the Neotropical zone in Central/South America [53]. According to the Bioregions 2023 Framework (https://www.oneearth.org/bioregions/), North American photographs represent samples from numerous biogeographic subrealms including the North Pacific Coast, American West, Mexican Drylands, Great Plains and the Southeast U.S. Savannas and Forests. Photographs of caracaras from Central America include the Central America and Caribbean biogeographic subrealms, with those from South America spanning the Andes and Pacific Coast, Upper South America, Amazonia, Brazilian Cerrado and Atlantic Coast and the South American Grassland subrealms.” (L120-132).

Section “Web-sourced…”.

If you decide to keep only with the Macaulay Library data, you will have to explain why you did not included searches in the WikiAves, iNaturalist and other databases. Like tell that you sample is large, huge, when compared with those of previous studies. On the other hand, if you decide to include other databases, then add text to explain how you did the searches.

>>> We have now included additional data from iNaturalist, please see response to comment above relating to this.

Line 116. You repeated…why not use “For them….” ?

>>> ‘useable’ now replaced with ‘For them’ (L142-143).

Line 117. What did you consider (adult or non-adult) when you could see an adult with a prey near a nestling in the nest ? This type of photograph has your two types of caracaras….

>>> The editor raises a very good point here relating to photographs where more than one bird (and age group) were visible in the photograph. We believe this was only the case for a handful of photographs. When this occurred, we only extracted age data from the bird that was interacting with the food item. We have clarified this in-text to benefit the reader 

“For photographs that contained more than one caracara of each age group, we only extracted data for the individual interacting with the prey item” (L145-147).

Line 188. I did not understand item 4 (location)…can you briefly explain within brackets ?

>>> We assume the editor refers to “location” on line 120 of the original submission and not line 188. By location we refer to the surrounding features of the environment within the wider photograph, e.g., type of fencing present, road signs, etc. We have revised the text to clarify this point:

“visual characteristics or landscape features within the photograph…” (L148-149).

Line 123. These brackets are too distant. Do you really need them ? Six lines within them…can you try to reduce this text (Lines 121-130)?

>>> We agree with the editor here that the use of brackets is not appropriate for the text. As such, we have removed the brackets and streamlined the text within the paragraph:

“However, this was not always the case as there were occasions where multiple recordists uploaded images of the same feeding event across different days. This was particularly the case if the food item was large in size and thus took longer to decompose. We were able to ascertain whether these were duplicates by making an assessment based on the spatial location of the feeding event, evidence of the food item included and features of the surrounding location in the photograph.” (L152-157).

Section “Prey and age”.

Line 141, if at least two authors, than should be more than CP and FB…please check the writing.

>>> Thank you for highlighting this. We have revised to:

“visually assessed by at least two of the authors (C.P., F.B. or V.N.).” (L170).

Line 142. Better if you start a new paragraph with “Adult caracaras…”.

>>> New paragraph created as suggested (L172).

Next lines. It is not clear if you included nestlings or not. It appears that you included only juveniles outside nests, and adults. But some photographs might show 1-2 adults in nests with prey and nestlings, as I could note in WikiAves. Please clarify it.

>>>

Section “Population”.

Lines 152 and 154. Extended or extends?

>>> “extended” replaced with “extends” (L182).

You could divide this section in 2-3 paragraphs. It is too long.

>>> New paragraph created after “In an attempt…” (L199).

Calling a figure with a map would be welcome here.

>>> We have cited Figure 1 here, which show the geographic spread of the data set graphically and in a map format:

“We plotted all useable photographs in QGIS version 3.14.16 [59] and conducted an overlap analysis between the geo-referenced photograph data points and each population range polygon (Fig. 1a-b).” (LXX).

Section “statistical Analysis”

How did you control for potential influence of seasonality on these results ? For example, you might have more photographs in the rainy or dry season for a given caracara population… please note that Reviewer 1 has concern about this, or similar potential influence on data.

>>> The editor raises an interesting point here in relation to temporal differences in caracara diet. There are certainly challenges associated with studying temporal changes in raptor diets across both the northern and southern hemisphere, however, it is possible with some additional analyses. However, we believe that an additional temporal analysis would be beyond the scope of the current study given multiple aspects of the species’ diet presented here, e.g., 1) overall diet composition, 2) age effects on diet and 3) diet variation across a latitudinal gradient. However, this certainly warrants further investigation in the form of an additional study where we have the space and opportunity to conduct a more thorough assessment of the species’ diet throughout the year, without being restricted by other results presented here.

Line 238. You have to place Table 1 here.

>>> Table 1 has been moved to the position suggested by the editor (L271).

Results.

Line 244. As you write “Of these…”, this percentage would be 1501/2454). No ?

>>> Absolutely and thank you for picking this up. We have revised the percentage to “68.9%” based on 2381/3451*100. (L286-287).

Lines 250. You have to call “(Fig 1)”, abbreviated with no dot.

>>> Revised throughout manuscript.

In the end of this paragraph (Line 258), you have to bring the captions of Fig 1 and Fig 2 (but not the figures, that should be kept in the end of the manuscript). Please check Instructions again.

>>> Full figures positioned at the end of the manuscript, captions in-text (L303-322).

Line 277. Bring Table 2 here.

>>> There is no Table 2 within the main text of the manuscript.

Line 287. Bring here: Caption of figure 3…

>>> Caption for Fig 3 placed in text where suggested (L351-355).

Discussion

It would be easier to read if you use subtitles for sections of the Discussion.

>>> Thank you for this suggestion, we have now included the following subtitles to help guide the reader through the Discussion section:

1. Crested Caracara diet varied along latitudinal gradients (L367).

2. Comparisons between other diet studies (L386).

3. Web-sourced photography is a useful tool to study raptor diets across large spatial scales (L402-403).

4. Study Limitations (L424).

5. Utilizing citizen science data can be a time- and cost-effective method to study raptor diets (L452-453).

6. Conclusions (L471).

Make sure that you discuss more extensively with previous study on the diet of the species, influence or not of age among raptors, influence of latitude on raptor diets. Your current discussion is a bit poor relative to these topics. Try to the emphasis on methodology.

Always consider an international readership by citing studies conducted in several ecoregions, countries, involving several species.

>>> We have now discussed our results in line with similarly, previous research on other raptor species: 

“For example, a global examination of Western Barn Owl (Tyto alba) and American Barn Owl (T. furcata) diets found a positive relationship in the proportion of mammal prey in colder environments [25]. Unlike previous studies which studied the diets of Eurasian Sparrowhawk and Martial Eagles [27,29], we did not find any age effects in the diet of caracaras which may be explained by their generalist nature and tendency to feed on carrion [47,66]. Eurasian Sparrowhawk are avian specialists [27,28] and age-related differences in diet may be more pronounced. Similar to our findings, a continental assessment of Montagu’s Harrier (Circus pygargus) diet found that mammalian prey increased at higher latitudes due to the prevalence of agricultural land cover at more northern latitudes [49].” (L408-417).

Both reviewers made some comments that might be useful for you here in the Discussion. For example, R2 commented that this study can be considered an example for the study less common, or more “difficult” species.

>>> This is an excellent point, which we agree with fully. As such, we have revised the text to reflect the comments from the editor and reviewer 2:

“Assuming that the study species is well-sampled, this approach can be applied to many raptor species, especially those that are less common or more challenging to study in the field,” (L466-468).

References.

The correct is to use like this “2. Smith TC….” Instead, you used “[2] Smith TC….”.

>>> All reference numbers have been revised following the comments from the editor.

Page numbers. There is no space before them (after the volume and :).

>>> Spaces between colon and page numbers have been removed for all relevant references.

Page number. You have to use a long dash between then, not an hifen (-).

>>> All hyphens between page numbers have now been replaced with em dashes.

DOIs. You have to provide it for all references that have it. Follow the format: (http://doi.org/10.....).

>>> Revised.

Ref 6. The initials in capitals should not occur, except for the first word and names….check for all.

>>> We have checked for this issue in all references and included lower cases where appropriate in references 6 and 7. 

Ref 26. It occurs with a duplicate. You will need to change all these numbers here and in the text.

>>> Now revised.

Please also check Instructions for 

---

## [Editor Report · Decision Letter 1]

17 May 2024

Continental scale dietary patterns in a New World raptor using web-sourced photographs

PONE-D-23-42552R1

Dear Dr. Connor Panter,

We’re pleased to inform you that your manuscript has been judged scientifically suitable for publication and will be formally accepted for publication once it meets all outstanding technical requirements.

Kind regards,

Dárius Pukenis Tubelis, Ph.D.

Academic Editor

PLOS ONE

Additional Editor Comments:

Dear Dr Connor Panter,

Thank you for submitting the revised version of your submission PONE-D-23-42552.

I noted that you followed most suggestions provided by the three external reviewers and me. On the few occasions that you disagred, you convinced me with your answers.

Thus, I appreciated all answers to the reviewers and all changes done to this new revised version. The new version was substantially improved in several aspects.

This manuscript provides interesting data, and you properly discussed aspects of the methods and variation in the diet of birds across large scale latitudinal gradients.

Thus, I consider that this study reaches the standard expected for publication in PLOS ONE.

I suggest Acceptance, and that this manuscript goes through the final steps for publication.

In my last reading, I found a few minor things to be fixed by you and co-authors prior to or during the proofs corrections. They are shown below.

I believe it will be a great contribution to PLOS ONE.

Thank you for considering PLOS ONE as home of your study.

Dr Dárius Tubelis

PLOS ONE Editor

**Final things to fix**:

Major section titles. You are using ABSTRACT, INTRODUCTION ....with all in capitals. Instead, you have to use only the first letter in capital (Introduction)....please check this throughout.

Line 48. when you have two references, you have to add a space after the comma. e.g. [6, 7]...please check this along the text.

Line 62. when you have three or more consecutive refs, you have to use a long dash (not hyphen) to separate the numbers. Please check all.

Line 75. Invert the numbers. Smaller numbers come prior to larger ones.

Line 103. You forgot to add the iNaturalist search...please add.

Lines 224 and 226. I think the wright is "km" and not "kms".

Lin2 277. Add a dot point after Table 1.

Line 395. Delete 40. Then use 39-48, as they are in sequence.

Line 416. Place Tyto alba in italics.

Line 435. Can you use "explain partly" ? Maybe these perfer large prey.... Small prey are not cost-effective, I think. This obeserved preference for larger prey be, mostly, a natural aspect.

References.

You are using a very, very, very long dash to separate page numbers in articles. I hope this is the correct one. PLOS ONE people with check this and maybe contact you.

All references appear to be well formatted.

Well done.

Congratulations on your study!

Dárius

---

## [Editor Report · Acceptance letter]

28 May 2024

PONE-D-23-42552R1 

PLOS ONE

Dear Dr. Panter, 

I'm pleased to inform you that your manuscript has been deemed suitable for publication in PLOS ONE. Congratulations! Your manuscript is now being handed over to our production team.

Kind regards, 

on behalf of

Dr. Dárius Pukenis Tubelis 

Academic Editor

PLOS ONE